# Interface engineering of charge-transfer excitons in 2D lateral heterostructures

Roberto Rosati ●[1] ✉, Ioannis Paradisanos ●[2], Libai Huang ●[3], Ziyang Gan[4,5], Antony George ●[4,5], Kenji Watanabe ●[6], Takashi Taniguchi ●[7], Laurent Lombez[2], Pierre Renucci[2], Andrey Turchanin ●[4,5], Bernhard Urbaszek[2,8] & Ermin Malic[1]

The existence of bound charge transfer (CT) excitons at the interface of monolayer lateral heterojunctions has been debated in literature, but contrary to the case of interlayer excitons in vertical heterostructure their observation still has to be confirmed. Here, we present a microscopic study investigating signatures of bound CT excitons in photoluminescence spectra at the interface of hBN-encapsulated lateral $MoSe_2$-$WSe_2$ heterostructures. Based on a fully microscopic and material-specific theory, we reveal the many-particle processes behind the formation of CT excitons and how they can be tuned via interface- and dielectric engineering. For junction widths smaller than the Coulomb-induced Bohr radius we predict the appearance of a low-energy CT exciton. The theoretical prediction is compared with experimental low-temperature photoluminescence measurements showing emission in the bound CT excitons energy range. We show that for hBN-encapsulated heterostructures, CT excitons exhibit small binding energies of just a few tens meV and at the same time large dipole moments, making them promising materials for optoelectronic applications (benefiting from an efficient exciton dissociation and fast dipole-driven exciton propagation). Our joint theory-experiment study presents a significant step towards a microscopic understanding of optical properties of technologically promising 2D lateral heterostructures.

Monolayers of transition metal dichalcogenides (TMD) have attracted much attention due to their remarkable excitonic and optical properties[1,2]. So far the research has focused on vertical TMD heterostructures obtained by stacking TMD monolayers on top of each other[3]. These are characterized by spatially separated interlayer excitons forming an out-of-plane dipole and thus allowing, e.g., a gate-controllable exciton transport[4,5]. In comparison, much less is known about *lateral* TMD heterostructures[6–14], where two different

TMD monolayer materials are grown sequentially and covalently bond in the plane[6–12] (Fig. 1a). These structures show regular monolayer optics and transport features when optically excited far from the interface[12,13]. At the interface, however, bound charge transfer (CT) excitons have been theoretically predicted[15]. Here, the Coulomb interaction binds together electrons and holes that are spatially separated at opposite sides of the junction (cf. Fig. 1a, b). This spatial separation results in an in-plane dipole that is typically larger than in

---

[1]Department of Physics, Philipps-Universität Marburg, Renthof 7, D-35032 Marburg, Germany. [2]Université de Toulouse, INSA-CNRS-UPS, LPCNO, 135 Avenue Rangueil, 31077 Toulouse, France. [3]Department of Chemistry, Purdue University, West Lafayette, IN, USA. [4]Friedrich Schiller University Jena, Institute of Physical Chemistry, 07743 Jena, Germany. [5]Abbe Centre of Photonics, 07745 Jena, Germany. [6]Research Center for Functional Materials, National Institute for Materials Science, 1-1 Namiki, Tsukuba 305-0044, Japan. [7]International Center for Materials Nanoarchitectonics, National Institute for Materials Science, 1-1 Namiki, Tsukuba 305-0044, Japan. [8]Institute of Condensed Matter Physics, Technische Universität Darmstadt, 64289 Darmstadt, Germany. ✉e-mail: rosatir@staff.uni-marburg.de

 1

vertical heterostructures[15], where the dipole is limited by layer separation. Therefore, the CT exciton binding energy is expected to be smaller compared to interlayer excitons[15–17]. Furthermore, smaller band offsets have been predicted for lateral heterostructures[18], suggesting that CT excitons are expected to be energetically close to the intralayer excitons. This reduced energy separation from the bright-exciton energy makes their detection challenging and could explain that so far there have been no clear experimental signatures for the existence of bound CT excitons in lateral TMD heterostructures - in contrast to interlayer excitons in vertical heterostructures[3].

In this work, we develop a fully microscopic and material-specific many-particle theory to shed light on the existence of CT excitons in lateral TMD heterostructures. We also perform cryogenic photoluminescence (PL) measurements to directly check the theoretical predictions. Motivated by the recent progress in the growth of lateral heterostructures with atomically sharp interfaces[9,14,19–21], we theoretically investigate optimal conditions to find CT excitons (i) via interface engineering (interface widths, band offsets) and (ii) dielectric engineering (surrounding substrates). In particular, we address the competition between Coulomb-induced spatial confinement of excitons (Bohr radius) and interface widths. Considering the exemplary case of hBN-encapsulated MoSe$_2$–WSe$_2$ lateral heterostructures[14,20], we predict for small junction widths and low temperatures the appearance of an additional low-energy resonance in PL spectra that we assign to a bound CT exciton. To test this, we perform cryogenic PL measurements in hBN-encapsulated MoSe$_2$–WSe$_2$ lateral heterostructures with a high-quality, very narrow junction width of ~2–3 nm[14]. We find PL emission peaks at the heterojunction in the high-quality samples that are below the MoSe$_2$ and WSe$_2$ intralayer excitons and that present a strong indication for the bound CT excitons predicted by our microscopic theory. Our joint theory–experiment study presents an important advance for a microscopic understanding of lateral TMD heterostructures, as we identify key conditions for the observation of CT excitons in terms of interface and dielectric engineering. Furthermore, we predict CT exciton binding energies of just a few tens of meV as well as extraordinarily large dipole moments for hBN-encapsulated materials. This indicates that lateral heterostructures with ultrathin junctions and weakly bound CT excitons to have also technological relevance for optoelectronic devices due to the expected high exciton mobility[13], efficient exciton dissociation, and diode-like exciton transport across the interface[14].

## Results

We investigate the exemplary case of an hBN-encapsulated MoSe$_2$–WSe$_2$ lateral heterostructure. We start with our microscopic theory and compare then with our cryogenic PL measurements. Figure 1b schematically shows the spatial variation of single-particle energies $E_{Mo/W}^0(x)$ in the considered lateral heterostructure. The conduction and valence bands form offsets $\Delta E_c, \Delta E_v$ at the interface, typically inducing a type II alignment[12,20,22–24] with the conduction band minimum located in the MoSe$_2$ layer[18]. Note that for gate-induced homojunctions[25–27], the band offsets are the same, i.e. $\Delta E_c = \Delta E_v$, potentially leading to bound excitons for p–i–n junctions confined to a few tens of nanometers[28]. At the interface, CT excitons can be built (purple oval) with the minimum continuum energy $E_{CT}^0 = E_{Mo}^0 - \Delta E_v = E_W^0 - \Delta E_c$. Here, we focus on bright CT excitons with the hole located at the K valley in the WSe$_2$ layer and the electron located at the K valley in the MoSe$_2$ layer, as this is the energetically lowest CT configuration, cf. the Supplementary material. Dark CT excitons could be important e.g. in lateral WSe$_2$–WS$_2$ heterostructures, where the minimum of the conduction band is located in the WS$_2$ layer[18]. Importantly, this CT continuum is lower in energy than monolayer bandgaps suggesting a high occupation of these states. To obtain the energy of bound CT excitons, Coulomb interaction

needs to be included resulting in excitonic energies (cf. Fig. 1c, d). The two-dimensional nature of TMD monolayers induces a reduced screening of the Coulomb interaction. The weakly screened Coulomb attraction leads in monolayers to quantization in the relative coordinate resulting in the formation of Coulomb-bound electron–hole pairs (excitons) $X_{Mo/W} = E_{Mo/W}^0 - X_{Mo/W}^b$ with large exciton binding energies $X_{Mo/W}^b$. The lowest 1s excitons are characterized by a Bohr radius in the range of one nanometer for hBN-encapsulated TMD monolayers[29]. At the interface of a lateral heterostructure an additional quantization of the center-of-mass motion can occur. The Coulomb-induced binding of spatially separated electrons and holes can form bound CT excitons that are localized at the interface with the energy $X_{CT} = E_{CT}^0 - X_{CT}^b$. However, their binding energies $X_{CT}^b$ are expected to be smaller than in the intralayer case due to the spatial separation between electrons and holes. This reduced binding energy for spatially-separated excitons is qualitatively similar to interlayer excitons in vertical TMD heterostructures[16,17], however, the separation of the latter is limited to the interlayer distance of the two TMD layers (although extendable via spacers[5]). In contrast, the separation of electrons and holes in a CT exciton is not limited by any geometrical constraint and can be principally much larger[13,15]. As a direct consequence, CT excitons are expected to have smaller binding energies compared to interlayer excitons, but exhibiting a large static electric dipole (cf. the Supplementary Materials). One important goal of this work is to study under what conditions these bound CT excitons $X_{CT}$ can be observed, i.e. when are they clearly below the $X_{Mo}$ exciton and have a sufficiently large oscillator strength. To reach this goal we perform interface and dielectric engineering in our calculations, allowing us to shift the relative position of intralayer and CT excitons (cf. Fig. 1c, d).

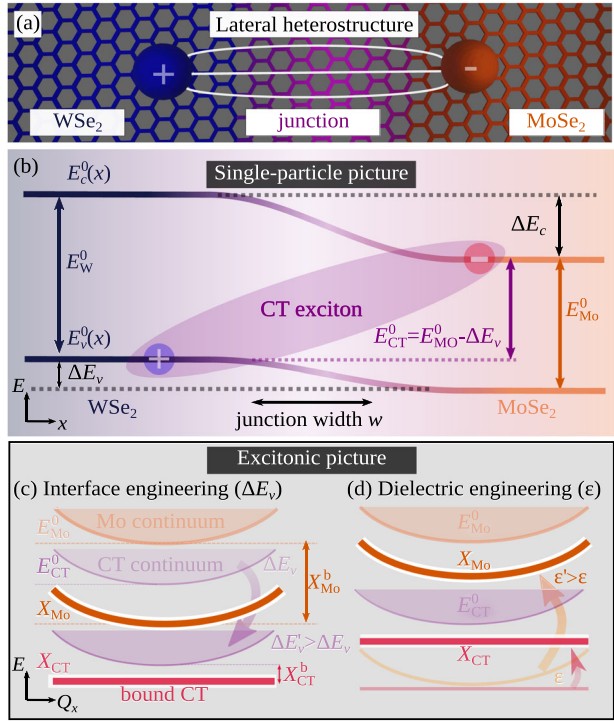

**Fig. 1 | Lateral heterostructures. a** Two TMD monolayers (e.g. MoSe$_2$ and WSe$_2$) are stitched laterally. **b** They have intrinsic bandgaps $E_{Mo}^0$ and $E_W^0$ while forming conduction and valence band offsets $\Delta E_c, \Delta E_v$ around the junction. Spatially separated electrons and holes across the interface form charge-transfer (CT) excitons with the corresponding continuum energy $E_{CT}^0 = E_{Mo}^0 - \Delta E_v$. **c, d** Bound CT excitons $X_{CT}$ (red flat line) appear below the energy of intralayer MoSe$_2$ exciton $X_{Mo}$ (orange line) for either large band offsets $\Delta E_v$ (interface engineering) or large dielectric constants $\varepsilon$ (dielectric engineering).

## Methodology and key quantities

To describe the spatially dependent energy landscape in a lateral heterostructure, it is crucial to include both the material-specific single-particle energies (Fig. 1b) as well as the Coulomb interaction that forms excitons (Fig. 1c, d). To this purpose, we investigate the excitonic eigenstates $|\Psi_n\rangle$ with eigenenergies $E_n$ of the Schrödinger equation $H|\Psi_n\rangle = E_n|\Psi_n\rangle$ with the Hamilton operator $H$ including both the spatially dependent single-particle energies $E_\lambda(x)$ (with the band index $\lambda = c, v$) and the Coulomb interaction between electrons and holes by using a generalized Keldysh potential $V_C(\mathbf{r})$[30,31]. Here $\mathbf{r}$ is the in-plane position vector, with $x$ and $y$ denoting the component perpendicular and parallel to the interface, respectively. Exploiting the symmetry along the $y$ direction parallel to the interface and the fact that the total exciton mass $M = m_e + m_h$ is much larger than the reduced mass $\mu = m_e m_h/(m_e + m_h)$, we can separate the eigenstates in a center-of-mass and a relative part with $\Psi_n(\mathbf{R}, \mathbf{r}) = \psi_n(R_x) e^{iQ_y R_y} \phi^{R_x}(\mathbf{r})$ with $\mathbf{r}$ as the relative coordinate and $\mathbf{R}$ and $\mathbf{Q}$ as the center-of-mass coordinate and momentum, respectively[15]. Here, $\phi^{R_x}(\mathbf{r})$ and $\psi_n(R_x)$ are the solutions of the corresponding Schrödinger equations for the relative and the center-of-mass motion:

$$\left[\frac{-\hbar^2 \nabla_\mathbf{r}^2}{2\mu} + V_C(\mathbf{r}) + V^{R_x}(\mathbf{r})\right] \phi_i^{R_x}(\mathbf{r}) = \tilde{E}_i(R_x)\phi_i^{R_x}(\mathbf{r}), \quad (1)$$

$$\left[-\frac{\hbar^2}{2M}\partial_{R_x}^2 + \tilde{E}_i(R_x)\right] \psi_{n,i}(R_x) = E_{n,i}\psi_{n,i}(R_x), \quad (2)$$

where $V^{R_x}(\mathbf{r}) = E_c^0(\mathbf{r}, R_x) - E_v^0(\mathbf{r}, R_x)$ acts as an interface potential given by the space-dependent band edges $E_{c,v}^0$. Note that the quantum numbers $n$ and $i$ describe the quantization in the center-of-mass and relative motion, respectively. In this work, we focus on the energetically lowest excitons corresponding to the $i = 1s$ states. In the case without a junction, there are no band offsets, i.e. $\Delta E_{c/v} = 0$ in Fig. 1b, and Eq. (1) becomes the well-known Wannier equation with a space-independent potential and $\tilde{E}_i(R_x) \equiv X_i$. In this limit, the center-of-mass equation (Eq. (2)) becomes trivial corresponding to fully delocalized plane waves $\psi_n(R_x) \equiv e^{iQ_x R_x}$ and resulting in $E_{n,i} \equiv E_{Q_x,i} = X_i + \hbar^2 Q_x^2/2M$. This implies that the center-of-mass motion of excitons is free and there is no quantization.

Solving Eqs. (1) and (2), two distinct situations can occur for the ground-state energy $E_0$, i.e. either (i) $E_0 = X_{Mo}$ or (ii) $E_0 < X_{Mo}$. In the first case, the regular MoSe$_2$ 1s exciton is the lowest state and is expected to dominate the optical response. In the latter case, the CT exciton $E_0 \equiv X_{CT}$ is the lowest state and could be principally observed in optical spectra. These CT states can be both bound or unbound and they are separated by the corresponding exciton binding energy $X_{CT}^b$, cf. the red and purple lines in Fig. 1c. The conditions for the visibility of the bound CT excitons are a relatively large binding energy (higher than thermal energy to prevent thermal dissociation into unbound states) and that the state is located clearly below the lowest intralayer exciton ($X_{Mo}$ for the investigated structure) and thus carrying a sufficiently large occupation.

To optimize the visibility of CT excitons in experiments we need to meet two conditions: (i) sufficiently low temperatures to avoid thermal dissociation of CT excitons and (ii) high sample quality so that the $X_{Mo}$–$X_{CT}$ energy separation is larger than the optical transition linewidth. Note that we recently reported high structural (electron microscopy) and optical quality (exciton transport) at the junction in CVD-grown MoSe$_2$–WSe$_2$[14]. The MoSe$_2$–WSe$_2$ lateral heterostructure offers a small lattice mismatch between MoSe$_2$ and WSe$_2$, while encapsulation of the samples with hBN minimizes the dielectric disorder[32] and promotes the intrinsic optical properties of the material in experiments performed at a temperature of $T = 4$ K. In addition, we

will show below that hBN-encapsulation plays an important role for CT exciton optics.

## Charge-transfer excitons

To determine the exciton energy landscape, we solve the Schrödinger equation (Eqs. (1) and (2)). We consider hBN-encapsulated samples and start with studying the limit of a relatively small band offset of $\Delta E_v = 100$ meV. Here, the energetically deepest excitons are found to be $X_{Mo}$ states (cf. Fig. 2a). Momentum-dark exciton states have been neglected, as they are energetically higher than the monolayer exciton $X_{Mo}$ (cf. the Supplementary material). The corresponding $X_W$ states are located 70 meV above, reflecting the band gap difference of MoSe$_2$ and WSe$_2$ (Fig. 2a). The center-of-mass dispersion is characterized by a parabola, and their wavefunctions $\psi(R_x)$ are confined either on the right- or on the left-hand side of the interface. For small band offsets, the binding energy of monolayer excitons is stronger than the band offset. As a result, we find no bound CT excitons as the energy of the CT continuum is much higher than the intralayer exciton energy $X_{Mo}$ (cf. Fig. 1).

The energy landscape changes significantly, when we increase the band offset to $\Delta E_v = 215$ meV, which is a realistic value for lateral TMD heterostructures[13,18]. Interestingly, we find bound CT excitons to be the lowest states (cf. Fig. 2b). They have a flat dispersion indicating localization of excitons, or to put it in other words, there is a quantization of the center-of-mass motion across the junction. These CT exciton states are unquantized along the interface, i.e. forming a one-dimensional CT-exciton channel. We predict two bound CT states and plot their center-of-mass wave functions in Fig. 2c. These are broad in momentum space reflecting a localization in real space around the interface and induced by the Coulomb attraction between the spatially separated electrons and holes. This is in strong contrast to the case of a regular monolayer without a junction, where the center of mass motion is free and the wave functions are very narrow in momentum space and fully delocalized in real space.

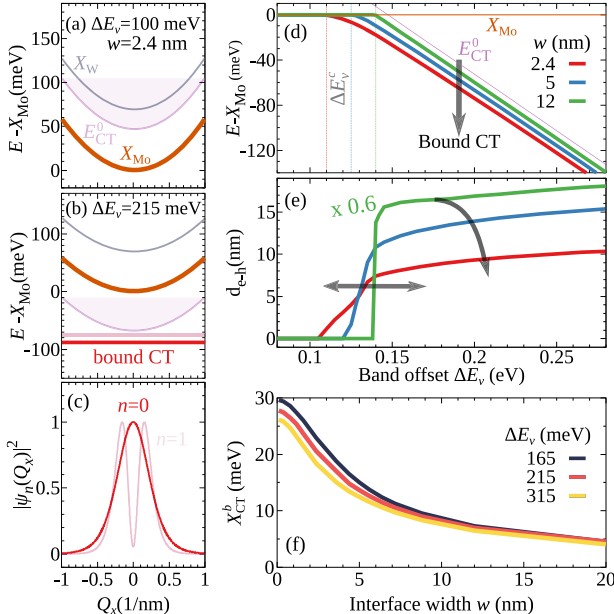

**Fig. 2 | Interface engineering. a** and **b** Dispersion relation of an hBN-encapsulated MoSe$_2$-WSe$_2$ lateral heterostructure for $\Delta E_v = 100$ and 215 meV, respectively. **c** Wave function of the two lowest bound CT excitons for $\Delta E_v = 215$ meV. **d** The energy of the lowest CT exciton (relative to $X_{Mo}$) and **e** its in-plane dipole $d_{e-h}$ revealing the appearance of bound CT excitons for band offsets larger than -100 meV. **f** CT exciton binding energy as a function of junction width $w$ for three different band offset values, revealing that sharp interfaces allow deeply bound CT excitons with $X_{CT}^b \approx 30$ meV.

We find that the bound states have typically an alternating symmetry, resulting in states with a finite and a negligible component in $Q_x = 0$, respectively (Fig. 2c). The vanishing $Q_x = 0$ component has a direct consequence for their oscillator strength so that only even states can emit light. The oscillator strength is also affected by the relative wavefunction, as the radiative recombination rate is proportional to the probability $|\phi(\mathbf{r} = 0)|^2$ of finding electrons and holes in the same position[15] (cf. "Methods" section). Due to the large spatial separation, this is smaller by a factor of almost 35 for CT excitons compared to intralayer states in the situation studied in Fig. 2b. However, being the energetically lowest states, their higher occupation, in particular at low temperatures, could still compensate their smaller oscillator strength and make them visible in optical spectra. In addition, relatively large binding energies are important because they give rise to a larger oscillator strength by increasing $|\phi(\mathbf{r} = 0)|^2$ as the electron–hole separation is reduced.

To sum up, the crucial conditions for the visibility of bound CT states are that they have a relatively large oscillator strength and that they are considerably deeper in energy than the lowest intralayer exciton state. In the following, we investigate interface engineering (variation of band offset and junction width) and dielectric engineering (variation of substrates) to predict optimal conditions for experimental observation of bound CT excitons that have not been demonstrated so far.

## Interface engineering

Here, we investigate how the CT exciton energy, its in-plane dipole, and the binding energy depend on the band offset $\Delta E_v$ and the interface width $w$ (Fig. 1). Note that varying $\Delta E_c$ gives qualitatively the same results. The band offset can be engineered by growing lateral heterojunctions of different TMD monolayers. The junction width $w$ depends on the exact growth technique and conditions. Recently, there has been an impressive technological development in lateral heterostructures allowing the realization of atomically narrow junctions of just a few nanometers[13,14,20,23,24,33,34]. In Fig. 2d, e we show the energetically lowest exciton state $E_0$ and its in-plane dipole $d_{e-h}$, respectively. To this end, we solve the Schrödinger equation (Eqs. (1) and (2)) as a function of the band offset $\Delta E_v$ for three different junction widths $w = 2.4$, 5, and 12 nm. The lower values correspond to recent experimentally realized sharp interfaces[13,14,20,24]. Importantly, our calculations show that for band offsets smaller than a critical value of about 100 meV, there are no bound CT excitons, but rather the regular MoSe$_2$ exciton $X_{Mo}$ is the lowest state (orange line). Increasing $\Delta E_v$, we observe that after a width-dependent critical value (defined as $\Delta E_v^c$) bound CT excitons become the lowest states with linearly increasing separation from $X_{Mo}$ as $\Delta E_v$ becomes larger. A similar behaviour is predicted for the free-standing case, but with a much larger $\Delta E_v^c \approx 200$ meV (cf. the Supplementary material). The binding energy $X_{CT}^b$ is enhanced for smaller junction widths $w$ (i.e. the red curve is further away from the purple curve in Fig. 2d).

To understand this, we plot the CT binding energy as a function of the junction width (Fig. 2f) for three different values of $\Delta E_v > \Delta E_v^c$. We find that for $\Delta E_v = 165$ meV the binding energy decreases by a factor of 3 when going from $w = 2.4$ to $w = 12$ nm ($X_{CT}^b \approx 23$ and 7 meV, respectively). Importantly, only for narrow junction widths, we predict binding energies of the order of the thermal energy also at room temperature. CT excitons with lower binding energy are thermally unstable and are expected to quickly dissociate into continuum states[35]. In addition, lower binding energies result in a smaller oscillator strength via a reduction of $|\phi(\mathbf{r} = 0)|^2$. We also observe that the binding energy is nearly independent of the band offset (almost overlapping lines in Fig. 2f), in particular for offsets $\Delta E_v$ much larger than the critical one. For offsets just larger than $\Delta E_v^c$, we predict a monotonic decrease of $X_{CT}^b$ with increasing $\Delta E_v$[15] (cf. the Supplementary material). As a consequence, the energy of the bound CT excitons $X_{CT}$ directly follows

the linear decrease of the CT continuum energy (purple line in Fig. 2d) as a function of the band offset.

The abrupt reduction of the CT exciton binding energy for increasing the junction width $w$ (Fig. 2f) induces an increase of the critical band offset $\Delta E_v^c$ from approximately 110–140 meV for junction widths $w$ going from 2.4 to 12 nm (cf. the critical values in Fig. 2d). For the case of $\Delta E_v = X_{Mo}^b$ the energy of the MoSe$_2$ exciton $X_{Mo}$ exactly coincides with the energy of continuum states $E_{CT}^0$ (cf. Fig. 1). For the general case, the critical band offset has to be defined as $\Delta E_v^c = X_{Mo}^b - X_{CT}^b$, such that the bound CT exciton becomes the energetically lowest state. As the binding energy of the monolayer exciton $X_{Mo}^b$ does not depend on the junction width, $X_{CT}^b$ is the crucial quantity. The latter has been shown to be very sensitive to the junction width (Fig. 2f). This explains why the critical band offset is increased for higher junction widths (i.e. smaller $X_{CT}^b$). This crucial dependence of $X_{CT}^b$ as a function of the junction width stems from the competition between the junction width $w$ and the Bohr radius $r_B$. The latter provides the spatial scale at which Coulomb-bound electrons and holes can redistribute around a center-of-mass position[29]. When $w \gg r_B$, excitons need huge dipoles $d_{e-h}$ for their electron/hole constituents to reach the energetically favourable spatial positions. As a result, bound CT excitons show very small binding energy. In the opposite case of $w \lesssim r_B$, the CT exciton experiences the maximum band offset already for small spatial separations resulting in large binding energies.

We now investigate the CT-exciton in-plane dipole $d_{e-h}$ as a function of the band offset (Fig. 2e). Similarly to the case of CT exciton energy in Fig. 2d, the dipole abruptly increases when the critical band offset $\Delta E_v^c$ is reached, i.e. when bound CT excitons are formed. For larger band offsets, the dipole only weakly increases. The dipole crucially depends on the binding energy of CT excitons: For larger $X_{CT}^b$ electrons and holes are bound close to the interface, i.e. they have a smaller in-plane distance and thus a smaller dipole. Since $X_{CT}^b$ depends strongly on $w$ and weakly on $\Delta E_v$ (Fig. 2f), there is only a small variation of $d_{e-h}$ with the band offset (above the critical value $\Delta E_v^c$), while $d_{e-h}$ increases by a factor of three for $w$ going from 2.4 to 12 nm ($d_{e-h} \approx 8$ and 27 nm, respectively, cf. red and green lines in Fig. 2e). The predicted dipoles are in the range of several nanometers, which is in good agreement with previous studies[13,15]. The values are much larger than for interlayer excitons in vertical heterostructures, where the electron–hole separation is limited by the layer distance[4]. The combination of small binding energies and large dipoles is attractive for optoelectronic applications due to efficient exciton dissociation and quick exciton propagation[13]. From the perspective of exciton optics, this can bring two limitations: First the larger $d_{e-h}$, the smaller is the binding energy $X_{CT}^b$ (Fig. 2e, f) and the less stable CT excitons are. Second, the increase of the dipole leads to a decrease of $|\phi(\mathbf{r} = 0)|^2$ resulting in a lowering of the oscillator strength with crucial implications for the visibility of CT excitons in experiments.

In a nutshell, by performing interface engineering one can achieve thermally stable bound CT excitons for atomically sharp interfaces. In particular, for the case of hBN-encapsulated MoSe$_2$–WSe$_2$, we predict binding energies of $X_{CT}^b \approx 20$–30 meV.

## Dielectric engineering

Besides interface engineering, Coulomb interaction can be changed by varying the dielectric environment determined by the substrate. We focus again on the lateral MoSe$_2$-WSe$_2$ heterostructure with a narrow junction width of $w = 2.4$ nm and a band offset of $\Delta_v = 215$ meV (i.e. above the critical value discussed in Fig. 2). These values are realistic according to the previous studies on lateral heterostructures[13,18]. Note that in our study we consider the band offset and the interface width to be robust with respect to the change in the dielectric environment.

In Fig. 3a we show the bound and unbound CT energies $X_{CT}$ and $E_{CT}^0$ as a function of the dielectric constant $\varepsilon$ of the substrate. We focus on CT energies relative to the intralayer MoSe$_2$ exciton $X_{Mo}$, as the

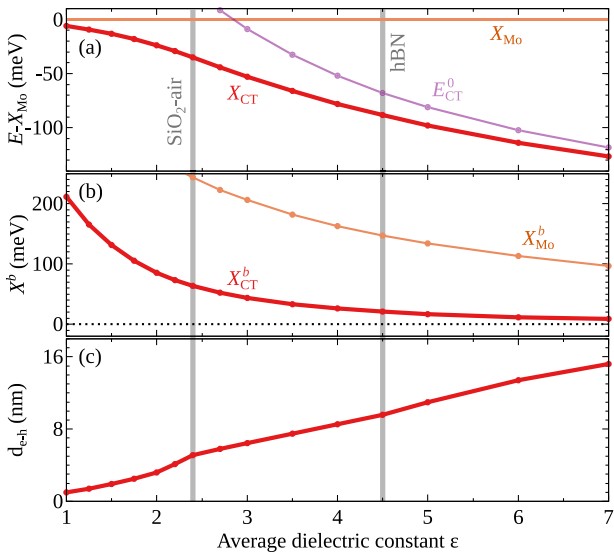

**Fig. 3 | Dielectric engineering. a** Bound and unbound CT exciton energies $X_{CT}$ and $E_{CT}^0$ (relative to the intralayer MoSe$_2$ exciton $X_{Mo}$), **b** CT binding energy $X_{CT}^b$, and **c** the corresponding CT exciton dipole $d_{e-h}$ as a function of the dielectric constant of the substrate (for the band offset $\Delta E_v = 0.215$ eV and the interface width $w = 2.4$ nm).

occupation of CT excitons is determined by their relative spectral distance to the monolayer exciton. We find a considerable shift to lower energies for increasing $\varepsilon$. The energy separation from $X_{Mo}$ of the bound CT exciton $X_{CT}$ is reduced from approximately 6 meV in the free-standing case ($\varepsilon = 1$) to 88 meV for hBN-encapsulated samples. A similar decrease is also found for the unbound CT state $E_{CT}^0$. As we are considering only relative energies (with respect to $X_{Mo}$), the dependence of the band gap energy $E_{Mo}^0$ on the dielectric screening is cancelled out. Thus the crucial quantities here are the binding energies $X_{Mo}^b$ and $X_{CT}^b$ of the monolayer and the bound CT excitons. The latter is very sensitive to the dielectric environment, as shown in Fig. 3b. In particular, the decrease of $X_{Mo}^b$ (orange in Fig. 3b) is responsible for the behaviour found for the energy of unbound CT excitons $E_{CT}^0$ (purple in Fig. 3a). Note that for $\varepsilon \approx 3$, the MoSe$_2$ exciton $X_{Mo}$ is shifted above the unbound CT energy resulting in a sign change in the purple line in Fig. 3a.

Interestingly, we predict a drastic decrease of $X_{CT}^b$ resulting in $X_{CT}^b$ being much smaller than $X_{Mo}^b$ (by approximately a factor of 4 and 7 in the case of SiO$_2$-air and hBN-encapsulation, respectively). This drastic decrease is in contrast to the situation in vertical heterostructures, where the binding energies are comparable for intra- and interlayer excitons[16,17]. This difference between vertical and lateral heterostructures can be ascribed to the much larger spatial electron-hole separations in CT excitons compared to interlayer excitons, where the separation is limited by the interlayer distance in vertical heterostructures. The CT binding energy decreases with the increasing dipole (cf. the Supplementary material), similar to the behaviour of interlayer excitons with increasing interlayer spacing[16]. Only for free-standing lateral heterostructures, we predict that CT excitons have dipoles $d_{e-h} \approx 1$ nm comparable with vertical heterostructures, resulting in comparable binding energies. In contrast, for substrates with an increasing dielectric constant, we find significantly enhanced in-plane dipole moments, e.g. $d_{e-h} \approx 5$ nm for the SiO$_2$ substrate or $d_{e-h} \approx 9.6$ nm for hBN-encapsulated samples. The increase in $d_{e-h}$ leads to a decrease in the CT binding energy as well as of the radiative recombination rate by one order of magnitude compared to the free-standing case.

The behaviour of $X_{CT}$ in Fig. 3a results from the non-trivial interplay of $X_{CT}^b$ and $X_{Mo}^b$. The CT exciton binding energy $X_{CT}^b$ decreases from about 200 meV in the free-standing case ($\varepsilon = 1$) to just a few meV

in the presence of high-dielectric substrates (Fig. 3b). As a consequence, bound and unbound CT energies almost coincide for large $\varepsilon$ (red and purple line in Fig. 3a). Furthermore, they shift well below the intralayer MoSe$_2$ energy $X_{Mo}$. In the limiting case of very large $\varepsilon$, the separation between $X_{CT}$ and $X_{Mo}$ tends toward the value of the band offset due to the negligible excitonic binding energies, (cf. Fig. 1). In the free-standing limit, we find $X_{CT} \approx X_{Mo}$, despite the large CT exciton binding energy. This occurs since for $\varepsilon \to 1$ also the unbound CT energy is shifted up relative to $X_{Mo}$ (purple line in Fig. 3a) and cancels out the change in $X_{CT}^b$, such that $X_{CT} = E_{CT}^0 - X_{CT}^b \approx X_{Mo}$. In this regime, the bound CT excitons are thermally stable thanks to binding energies of a few hundred meV (Fig. 3b), but they are located only slightly below $X_{Mo}$ (Fig. 3a). Thus, they are weakly populated and not visible in PL spectra, cf. the supplementary material. It is, however, in the intermediate range of $2 < \varepsilon < 5$ that one finds the optimal situation where we have a considerably large CT binding energy and at the same time the CT exciton is located well below the MoSe$_2$ exciton. For a SiO$_2$-air environment ($\varepsilon \approx 2.4$), we predict the CT exciton to be ~35 meV below $X_{Mo}$ with a binding energy of $X_{CT}^b \approx 63$ meV. The energy separation between CT and intralayer MoSe$_2$ excitons increases significantly in hBN-encapsulated heterostructures ($\varepsilon \approx 4.5$), but this comes at the price of a smaller binding energy of $X_{CT}^b \approx 20$ meV.

In a nutshell, high-dielectric substrates lead to bound CT excitons that are located much below the intralayer exciton and thus carry a large occupation, however, they are weakly bound and hence thermally unstable. The optimal case is reached for $\varepsilon \approx 2-5$ where we find CT excitons with a considerably large binding energy and still a sufficient occupation.

## Optical spectra

Now, we investigate whether bound CT excitons can be observed in photoluminescence spectra. First, we calculate a PL spectrum of an hBN-encapsulated MoSe$_2$-WSe$_2$ lateral heterostructures (with the band offset $\Delta E_v = 0.215$ eV and the interface width $w = 2.4$ nm) and then we perform cryogenic PL measurements. The starting point of our calculation is a focused laser excitation spot with an FWHM of 700 nm as in a typical experiment[36]. In a homogeneously excited system, the PL can be expressed by the Elliott formula describing the emission of bright exciton states[37]. In our case, this Elliott formula has to be extended to take into account the spatially confined laser excitation and excitonic states. To this purpose, we assume a Gaussian excitonic distribution $N(x_0, R_x)$ localized around $x_0 \equiv R_x^0$ with a spatial width $\Delta_x$ in agreement to the FWHM of the laser pulse. Here, $R_x$ is the exciton center-of-mass position. In the case without a junction, the spectral distribution is governed by the Boltzmann distribution. In the presence of a junction, however, the wavefunction of each state $|\psi_n\rangle$ plays an important role and determines the relative occupation of the state via a weight coefficient $c_n(R_x)$, i.e. $N_n(R_x) = N(x_0, R_x)c_n(R_x)$ (cf. "Methods" section for more details). This makes sure that we have a local thermal distribution. We generalize the Elliott formula for the PL intensity $I_n(E)$ of the state $|\psi_n\rangle$ taking into account that the laser pulse excites a spatially inhomogeneous exciton distribution. Thus, the spatially dependent PL reads after an optical excitation centered at $x_0$

$$I(x_0, E) = \sum_n I_n(E) \int dR_x\, c_n(R_x) N(x_0, R_x), \qquad (3)$$

i.e. we sum over all emitting states $|\psi_n\rangle$ and weight the emission by the coefficient $c_n(R_x)$. Note that we limit our study to momentum-direct radiative recombination since phonon sidebands are expected only in the WSe$_2$ but not in the MoSe$_2$ monolayer (as here momentum-dark excitons are not the energetically lowest states, cf. the Supplementary material)[38-40]. As a result, we also do not expect efficient indirect recombination of CT excitons as here the electron is located in the MoSe$_2$ layer (Fig. 1b). Furthermore, funneling effects[41] and exciton

thermalization/charge transfer effects[42,43] are beyond the scope of this work.

Now, we evaluate Eq. (3) and calculate spatially and spectrally dependent PL spectra at different temperatures for the hBN-encapsulated MoSe$_2$–WSe$_2$ lateral heterostructures. We tune the laser pulse position $x_0$ and fix the junction characteristics to values of $\Delta E_v = 0.215$ eV and $w = 2.4$ nm in accordance with predicted and measured values[14,18]. At moderate and high temperatures far away from the junction, we reproduce the regular monolayer PL spectrum and find the $X_{Mo}$ and $X_W$ excitons on the right-hand and the left-hand side, respectively (cf. Fig. 4a, b). When exciting at the interface, both features are still visible reflecting the large spatial width of the laser pulse (FWHM of 700 nm) that excites both sides of the heterojunction. Interestingly, when decreasing the temperature, a low-energy resonance appears approximately 90 meV below $X_{Mo}$ (cf. Fig. 4c, d). This can be clearly ascribed to the position of the CT exciton $X_{CT}$ (Fig. 2d). At low temperatures, CT excitons can result in a strong PL despite their low oscillator strength due to their large occupation as energetically lowest states. The PL emitted from CT excitons is particularly strong compared to $X_W$, as the bright exciton $X_W$ in the WSe$_2$ layer is higher in energy than $X_{Mo}$, and is thus only weakly populated. For the same reason, we find that $X_{Mo}$ is more intense than $X_W$ at low temperatures (Fig. 4c). Importantly, the new low-energy peak $X_{CT}$ is visible only in the presence of a narrow junction, (i.e. $w = 2.4$ nm), while it disappears for larger junction widths, as shown by the dashed orange line in Fig. 4d. This can be explained by the smaller spectral separation of CT excitons from the monolayer resonance at broader junctions (Fig. 2d), resulting in a smaller occupation of the CT state. In addition, the CT exciton

binding energy also considerably drops, and the electron–hole separation drastically increases (Fig. 2e). As a direct consequence, the radiative decay rate $\gamma_0$, which is given by the wavefunction overlap of electrons and holes (cf. "Methods" section), decreases by 4 orders of magnitude when moving from $w = 2.4$ nm to $w = 12$ nm.

To test the theoretical prediction we perform spatially dependent cryogenic PL measurements on the very same sample system, i.e. hBN-encapsulated MoSe$_2$–WSe$_2$. This sample set has shown high structural quality at the junction in electron microscopy and clear exciton transport from WSe$_2$ to MoSe$_2$ through the junction[14]. This allows us to show a direct comparison between theory and experiment (cf. Fig. 4d–f). In Fig. 4e, f we present the spectra from two different junctions. We find in the experiment a clear PL emission of about 80–100 meV below the $X_{Mo}$ resonance at several junctions. The emission in the CT-exciton energy range is absent far away from the junction (cf. thin bright blue line in Fig. 4e, f). This is in excellent agreement with the theoretical prediction (Fig. 4d) and is a strong indication of the direct emission from CT excitons. To further support this assignment, we have performed power-dependent studies, cf. the Supplementary material. The integrated intensity of the low-energy peak increases linearly with the excitation power, contrary to the saturating behaviour expected from defects[44,45]. In addition, we also observe a blue-shift of the peak with increasing excitation powers, similar to the behaviour of interlayer excitons, which blueshift due to dipole–dipole repulsion[46]. The observed shift could thus further confirm the dipolar origin of the low-energy peak.

By investigating samples in PL at cryogenic temperatures, the chances for the observation of the CT exciton are optimized also by the hBN encapsulation which, besides reducing the disorder (resulting in linewidth of less than 10 meV for $X_{Mo}$), leads to large energy separations between $X_{CT}$ and $X_{Mo}$ excitons, as explained in the dielectric engineering part of the manuscript (Fig. 3). We emphasize that both the narrow linewidth and the large-energy separation between $X_{CT}$ and $X_{Mo}$ are needed to observe CT excitons. The broader nature of the CT exciton in the experiment is likely to be related to sample imperfections or strain. A moderate red-shift of 20–30 meV of $X_W$ close to the junction[14] suggests the presence of strain that could result in an inhomogeneous broadening of $X_{CT}$, together with dielectric disorder and impurities. Furthermore, we observe trionic features resulting in multiple peaks around the energy of the MoSe$_2$ exciton that have not been taken into account in the theory. Note that PL emission at the calculated CT exciton energy has been observed in several junctions (cf. Fig. 4e, f). From a material perspective, it is likely that the junction width $w$ might vary for junctions grown on the same substrate. As a result, CT exciton formation does not necessarily occur at all junctions, due to the strong dependence on $w$ as shown in our calculations (Fig. 2f).

## Discussion

We have presented a joint theory–experiment study investigating the bound charge-transfer excitons at the interface of lateral two-dimensional heterostructures. We find in theory and experiment first signatures for the appearance of bound charge transfer excitons in cryogenic photoluminesce spectra of hBN-encapsulated lateral MoSe$_2$–WSe$_2$ heterostructures. We perform interface and dielectric engineering in our calculations and reveal critical conditions for the observation of charge transfer excitons including narrow junction widths (in the range of a few nm), relatively large band offsets (above 100 meV), and an intermediate dielectric screening ($\varepsilon \approx 2$–5). Our study provides novel insights into the characteristics of bound charge transfer excitons and will trigger future experimental and theoretical studies in the growing research field of lateral heterostructures. The latter also has a large technological potential as ultrathin junctions present quasi-one-dimensional channels with a strongly suppressed scattering with phonons and thus significantly enhanced exciton

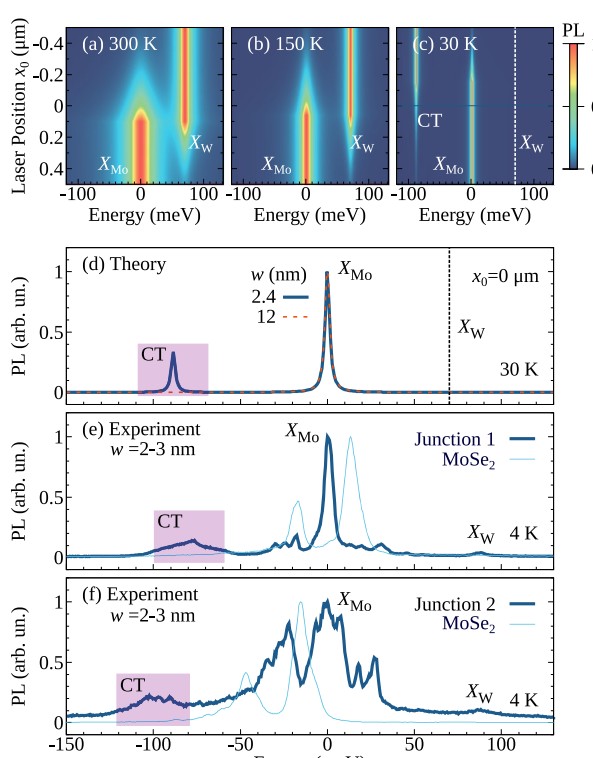

**Fig. 4 | Optical signatures.** Photoluminescence (PL) spectra of hBN-encapsulated MoSe$_2$–WSe$_2$ lateral heterostructures with the band offset $\Delta E_v = 0.215$ eV and the interface width $w = 2.4$ nm studied at **a** 300 K, **b** 150 K, and **c** 30 K. We excite the material with a laser excitation spot with an FWHM of 700 nm. **d** Cuts of the PL spectrum at the interface at 30 K. We also show the comparison to the larger interface width of 12 nm (dashed orange line). **e, f** Experimental PL spectrum at the junction and at MoSe$_2$ monolayer region, with two different interfaces considered. We find both in experiment and theory a low-energy resonance that we assign to a bound CT exciton.

mobility[47]. Additionally, the large intrinsic dipole of CT excitons is expected to lead to an efficient dipole–dipole repulsion that together with the 1D confinement could lead to excitonic highways as recently proposed[13]. A further key for technological application is exciton dissociation, i.e. the conversion of light absorption into electrical currents. Due to the huge excitonic binding energies of hundreds of meV, the charge separation is largely ineffective in TMD monolayers. In contrast, weakly bound CT excitons in lateral heterostructures will efficiently dissociate and thus facilitate charge separation. In our work, we show how to engineer lateral TMD heterostructures to obtain stable and highly dipolar CT excitons that have a high potential to boost exciton transport and exciton dissociation–both highly relevant for optoelectronic applications.

## Methods
### Microscopic modeling
To microscopically model charge transfer excitons in lateral TMD heterostructures, we solve the Schrödinger equation including the strong Coulomb interaction in TMD monolayers and the space-dependent dispersion relations induced by the junction (Fig. 1b). The Coulomb interaction $V_C(\mathbf{r})$ is described introducing a generalized Keldysh potential[30,31,48] for charges in a thin-film surrounded by a dielectric environment that is spatially homogeneous along the plane in terms of thickness and dielectric constant[30,31,48]. The in-plane variation of energy is described via spatially dependent single-particle energies $E_{c/v}^0(\mathbf{r})$ of electrons and holes, respectively. In particular, we take $E_{c/v}^0(\mathbf{r}) = \Delta E_{c/v}/2(1 - \tanh(4x/w)) + E_{Mo}^0(1 \pm 1)/2$, which recovers the situation in Fig. 1b[15]. The Schrödinger equation can be separated into equations for the relative and the center-of-mass motion (Eqs. (1) and (2)). We focus on electrons and holes located at the K valley in $MoSe_2$ and $WSe_2$, respectively, as all other CT electron–hole pairs are energetically higher, cf. the Supplementary material. While lateral heterostructures involving TMDs with different chalcogen atoms have a lattice mismatch, in $MoSe_2$–$WSe_2$ we can assume a strain-free interface[11,19]. Finally, we solve the coupled Eqs. (1) and (2) with space-independent $WSe_2$ electron masses[49] to obtain the eigenenergies and eigenfunctions, which in turn allow to determine the radiative recombination rate and the dipole $d_{e-h} \equiv |\mathbf{d}_{e-h}| = |\int dR_x d\mathbf{r} \mathbf{r} |\Psi(R_x, \mathbf{r})|^2|$. Note that finite dipoles are present only for CT states and only across the interface, i.e. $d_{e-h} = d_{e-h}^x$ and $d_{e-h}^y = 0$, where $d_{e-h}^{x/y}$ describes the component across and along the interface of $\mathbf{d}_{e-h}$, respectively.

To model the spatially dependent PL, we must take into account the junction in lateral heterostructures yielding the PL formula in Eq. (3). The appearing coefficients $c_n(R_x)$ can be obtained starting from the total center-of-mass excitonic distribution $N(\mathbf{R}) \propto \sum_{nn'} \langle \hat{X}_{n'}^\dagger \hat{X}_n \rangle \psi_{n'}^*(\mathbf{R}) \psi_n(\mathbf{R})$, where $\hat{X}_n^\dagger, \hat{X}_n$ are the creation/annihilation operators of an exciton in the state $n$ and where $\langle \hat{X}_{n'}^\dagger \hat{X}_n \rangle$ is the single-exciton density matrix. In the equilibrium of homogeneous low-density excitations, we find $\langle \hat{X}_{n'}^\dagger \hat{X}_n \rangle \equiv c e^{-E_n/k_B T} \delta_{nn'}$ with $c$ being the normalization constant reflecting the local density.

Applying such equilibrium condition to the general definition of $N(\mathbf{R})$ yields

$$N(\mathbf{R}) \equiv c \sum_n e^{-\frac{E_n}{k_B T}} |\psi_n(\mathbf{R})|^2 = N(\mathbf{R}) \sum_n c_n(\mathbf{R}) \quad (4)$$

with $c_n(\mathbf{R}) = e^{-\frac{E_n}{k_B T}} |\psi_n(\mathbf{R})|^2 \left[\sum_{n'} e^{-\frac{E_{n'}}{k_B T}} |\psi_{n'}(\mathbf{R})|^2\right]^{-1}$ providing the local occupation of state $|n\rangle$. In the monolayer limit one has $|\psi_n(\mathbf{R})|^2 = 1/A$ with $A$ being the area of the sample. Hence, the coefficients $c_n(\mathbf{R})$ become the normalized spatially independent Boltzmann distribution. A highly non-trivial dynamics is expected at the interface, where the charge transfer[42,43] into bound CT states is likely to lead to local features similar to those of phonon-induced carrier-capture[50,51]. In this work, we focus on stationary PL after exciton thermalization has occurred.

The space-independent PL of $I_n(E) = \tilde{\gamma}_n \frac{\tilde{\gamma}_n + \Gamma_n}{(E - E_n)^2 + (\tilde{\gamma}_n + \Gamma_n)^2}$ entering Eq. (3) describes the emission spectrum after radiative recombination of the state $|n\rangle$ according to the excitonic Elliott formula[37,38]. Phonon-assisted mechanisms are not included as they are expected to be strong only on the $WSe_2$ side of the junction but negligible for both $MoSe_2$ and CT excitons in the junction. The oscillator strength $\tilde{\gamma}_n = \gamma_n |\psi_{Q_x = 0}|^2$ is given by the product of the radiative rate $\gamma_n$ and the $Q_x = 0$ component of the squared wavefunction in center-of-mass momentum space $\psi_{Q_x}$ in view of the conservation of momentum after recombination into photons[52]. The radiative rate $\gamma_n$ can be extracted from the monolayer case[48] as $\gamma_n = \tilde{M} |\phi(\mathbf{r} = 0)|^2 / E_n$ with $\tilde{M}$ depending on the material and the substrate (via optical dipole moment or refractive index), while $E_n$ and $|\phi(\mathbf{r} = 0)|^2 = \int dR_x |\psi(R_x)|^2 |\phi^{R_x}(\mathbf{r} = 0)|^2$ are obtained from the solution of Eqs. (1) and (2), i.e. in particular including effects from the junction. While $\tilde{\gamma}_n$ determines the oscillator strength, i.e. the height of the resonances, $\Gamma_n$ describes the impact of exciton–phonon scattering on the shape of the resonances. As a full microscopic calculation of the latter including the junction is beyond the scope of this work, we estimate $\Gamma_n$ with the values obtained in the low-density limit for $MoSe_2$ and $WS_2$ monolayers[53].

### Sample fabrication and photoluminescence measurements
Our $MoSe_2$–$WSe_2$ lateral monolayer heterojunction is grown by chemical vapor deposition (CVD) synthesis that we reported recently[20]. For the hBN encapsulation we follow the water-assisted transfer method to pick up as-grown, chemical vapor deposition (CVD) lateral heterostructures using polydimethylsiloxane (PDMS) and deterministically transfer and encapsulate them in hBN[54,55]. Photoluminescence spectra are collected at $T = 4$ K in a closed-loop liquid helium (LHe) system. A 633 nm HeNe laser is used as an excitation source with a spot size diameter of $\approx 1 \mu$m and $6 \mu$W power, while in the Supplementary material we present results gradually increasing the power up to $25 \mu$W.

## Data availability
The datasets generated during and/or analysed during the current study are available from the corresponding authors on reasonable request.

## Code availability
The codes used to generate the data are available from the corresponding authors on reasonable request.

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

## Acknowledgements

We acknowledge funding from the Deutsche Forschungsgemeinschaft (DFG) via SFB 1083 and project 512604469 as well as from the European Union's Horizon 2020 research and innovation program under grant agreement no. 881603 (Graphene Flagship). Toulouse acknowledges partial funding from ANR IXTASE, Growth of hexagonal boron nitride crystals was supported by JSPS KAKENHI (Grants Nos. 19H05790, 20H00354, and 21H05233). Jena group financial support of the Deutsche Forschungsgemeinschaft (DFG) through CRC 1375 NOA (Project B2), SPP2244 (Project TU149/13-1), DFG grant TU149/16-1.

## Author contributions

R.R. and E.M. developed the theoretical model for CT-exciton engineering and optics. I.P., L.L., P.R., and B.U. devised and performed the PL measurements. Z.G., A.G., and A.T. grew the CVD samples. T.T. and K.W. grew the hBN bulk. All authors contributed to the writing of the manuscript.

## Funding

## Competing interests

The authors declare no competing interests.
