## [Peer Review File · Nature Communications]

Reviewers' Comments:

Reviewer #1:

Remarks to the Author:

The manuscript investigates the interface excitons in 2D lateral interface. This topic is not largely explored, due to the difficulty in the realization of high-quality lateral interface [6-12]. However, this basic research about microscopic understanding of optical properties could provide interesting application in optoelectronic devices (e.g. 1D confinement, charge separation).

Based on the idea reported in ref. 15, the authors predict low-energy CT exciton and a larger in-plane dipole with respect to interlayer exciton, induced by a spatial separation of few nanometres. Evidence of such exciton is expected in photoluminescence (PL) at low temperature, validated roughly by experimental results (in fig.4 the matching with the expected PL peak is not excellent. The dissimilarity is explained by sample imperfection and strain).

The work is well organized, but the novelty is compromised by ref. [15] for interface excitons and ref. [14] for optoelectronic characterization in lateral heterostructure. For this reason, the authors should emphasize the novelty highlighting: the improvement in the microscopic understanding introduced by the work with respect to other works (for instance ref [15]) and/or the improvement in the understanding of previous experimental data (for instance the measurements at room temperature of ref [14]).

Another critical point is the agreement between experiment and theory reported in fig. 4.

Specifically, the PL peak at ~ -100 meV, ascribed to interface exciton, exhibits a bandwidth larger than the theoretical prediction and a different peak position for the two junctions (fig. 4 e,f).

Moreover, the defects induce PL peaks below the absorption edge (DOI:

10.1021/acsaem.8b01561) and their appearances are most visible at low temperature (DOI:

10.1063/1.4729920). Assuming an inhomogeneous spatial organization in the lateral

heterostructure, the spatially localized PL peak (ascribed in the manuscript to an interface exciton) can be explained by a defect level. For instance, a similar peak around 1.45 eV is also observed in fig. 2b of ref. [14], but it cannot be ascribed to interface exciton because it is not localized in the lateral heterojunctions. The authors should validate the experiment/theory matching ruling out other easier explanations of PL peak.

Summing up, before the publication the authors should address in a more direct way the novelty introduced by the manuscript and they should support the experimental comparison more convincingly.

The numeric citations are referred to the manuscript references.

Reviewer #2:

Remarks to the Author:

In this study, Rosati and colleagues presented a joint theory-experiment study to investigate the bound charge-transfer excitons at the interface of lateral two-dimensional heterostructures. With a comprehensive calculations, they discussed how interface- and dielectric effects can modulate the exciton energy and binding energy of the charge-transfer exciton, and propose that the optimal conditions for an experimental observation of charge transfer excitons are the narrow junction widths (in the range of a few nm), relatively large band offsets (above 100 meV), and an intermediate dielectric screening ($\epsilon \approx 2 - 5$). Following the theoretical prediction, they observed a PL emission about 80 to 100 meV below the PL peak of MoSe₂ resonance in hBN-encapsulated MoSe₂-WSe₂ junctions, which is assigned as the charge-transfer exciton. These results sound reasonable and maybe useful for the further theoretical and experimental studies on lateral two-dimensional heterostructures. But there are a few questions that need to be clarified before I can recommend the paper for publication.

1, In Figure 2, the calculated binding energy of charge-transfer exciton is only about 30 meV in hBN-encapsulated MoSe₂-WSe₂ lateral heterostructure. While the binding energy of free standing TMD monolayers can be a few hundred meV in the previous studies (Ref.15 and GW-BSE calculations). Although I understand the dielectric screening can decrease the binding energy, the value presented here is too small. Maybe more comparisons with the previous experimental and theoretical results are needed.

2, In Figure 3, the calculate CT exciton dipole de-h can reach 16 nm, which is larger than the adopted interface width $w=2.4$ nm, can the author explain why?

3, In the previous experimental studies [Nano letters, 2015, 15(10): 6494-6500.], the WSe₂ monolayer exhibits an indirect quasi-particle gap with the conduction band minimum located at the Q-valley. While, in this study, the author assumes the conduction band minimum of WSe₂ is at K valley and take MoSe₂-WSe₂ as type II heterostructure. So is it possible that MoSe₂-WSe₂ is Type I lateral heterostructure, and that the observed PL emission below the PL peak of MoSe₂ resonance is from the WSe₂ indirect exciton, instead of the charge-transfer exciton?

Reviewer #3:

Remarks to the Author:

In the manuscript titled "Interface engineering of charge-transfer excitons in 2D lateral heterostructures", the authors report on an in-depth investigation of the electronic properties of charge transfer excitons, which are supported at the interfaces of lateral transition metal dichalcogenide (TMD) heterostructures. Due to the type II band alignment, these excitons are spatially indirect, alike their interlayer counterparts in vertical TMD heterostructures. Unlike the latter, however, the number of works available in the literature is comparatively small. The theoretical works focus mainly on the band alignment between the constituent materials by making use of ab-initio methods, while experimental works typically employ scanning probe microscopy. Those which use spectroscopic tools to unveil the properties of these heterostructures mainly do so at room temperature, presumably due to the relatively poor optical quality of the materials, usually grown by chemical vapour deposition. In this manuscript, the authors propose for the first time a fully microscopic theoretical investigation of the charge transfer exciton, they thoroughly explore their properties by (theoretically) performing dielectric and interface engineering. The manuscript is complemented by an interesting comparison between the simulated and experimental PL spectra, in which the authors identify a peak at energy lower than that of MoSe₂ exciton with the charge transfer exciton. This represents the first time this excitonic complex has been observed experimentally.

The manuscript is very well written. The authors guide the reader through their results very clearly. Both this and the topic make this manuscript of potential interest for the broad readership of Nature Communications. I would have a few points for the authors' consideration before my final recommendation.

1. In the "interface engineering" section, the authors discuss in detail how a few key quantities (e.g., energy and dipole of the charge transfer exciton). They however do not devote as much attention to the dependence of the binding energy on the band offset. Their simulation summarized in Fig 2f show that this quantity decreases for increasing band offset, which might be counterintuitive. I suggest to comment on this in the revised manuscript.

2. Can the author comment on the comparison of their simulated value of the binding energy of the suspended heterostructure ($\epsilon=1$) with that reported in PRB 98 115427, which seems to be considerably higher?

3. In the simulated spatial-dependent PL spectrum of fig 4c, the intensity of the charge transfer exciton seems to be larger on the WSe₂ side: could the authors clarify why this is the case?

4. At low temperature, always in fig 4c, one can note that a non-zero signal related to the exciton of MoSe₂ is present also relatively far from the interface: is this related to diffusion effects?

5. Did the authors perform some spatial dependence PL measurements at low temperature? In a prior publication, they made use of tip enhanced PL, which I suppose would not be possible to perform at cryogenic temperature, but perhaps a raster scan of the PL spot across the heterostructure could be interesting and show also experimentally the simulated spatial dependence of the charge transfer exciton.

6. How does the ratio of intensity between the (neutral) exciton of MoSe₂ and the charge transfer exciton compare in the simulated and experimental spectra?

Response letter to reviewers' comments

We thank the reviewers for carefully reading our manuscript and for providing a detailed feedback, including positive statements as well as constructive critique, which helped us to further improve the presentation of our work. We have addressed all comments and provide a detailed point-by-point response below. We have performed new calculations and new experiments and have created a supplementary material. Corresponding changes in the revised manuscript are marked in blue for clarity.

Reviewer: 1

1 Comment *“The manuscript investigates the interface excitons in 2D lateral interface. This topic is not largely explored, due to the difficulty in the realization of high-quality lateral interface [6-12]. However, this basic research about microscopic understanding of optical properties could provide interesting application in optoelectronic devices (e.g. 1D confinement, charge separation). Based on the idea reported in ref. 15, the authors predict low-energy CT exciton and a larger in-plane dipole with respect to interlayer exciton, induced by a spatial separation of few nanometres. Evidence of such exciton is expected in photoluminescence (PL) at low temperature, validated roughly by experimental results (in fig.4 the matching with the expected PL peak is not excellent. The dissimilarity is explained by sample imperfection and strain). The work is well organized, but the novelty is compromised by ref. [15] for interface excitons and ref. [14] for optoelectronic characterization in lateral heterostructure. For this reason, the authors should emphasizing the novelty highlighting: the improvement in the microscopic understanding introduced by the work with respect to other works (for instance ref [15]) and/or the improvement in the understanding of previous experimental data (for instance the measurements at room temperature of ref [14]).”*

Answer We thank the referee for acknowledging that the topic of our research is interesting and potentially important for technological applications. We are also thankful for raising the point of novelty that we are happy to clarify. While our microscopic description of charge-transfer (CT) excitons is based on the Born-Oppenheimer approximation, as proposed in Ref. [15], we go beyond and focus on exciton optics, which allows us to predict signatures of charge transfer excitons that are accessible in experiments. We build the bridge between the microscopic theory and experiment and on top and we also perform the experiment. In this perspective, our joint theory-experiment study goes considerably beyond Ref. [15]. In comparison to the purely experimental study of Ref. [14], our work has a clear focus on exciton optics providing for the first time signatures of charge transfer excitons in lateral heterostructures. This experimental measurement goes beyond the one reported in Ref. [14], which in contrast focused on transport properties by exploring the spatially dependent relative height of monolayer exciton peaks. The first-time optical investigation of charge transfer excitons is thus a key improvement with respect to both Refs. [14] and [15], as well as with respect to the current literature.

On top, we provide new microscopic insights into the importance of the substrate. We show that an hBN-encapsulation is crucial not only in terms of a reduced disorder (i.e. narrow spectral lines), but also in terms of dielectric impact on the Coulomb interaction. The energy separation between charge transfer and monolayer excitons drastically increases for a larger dielectric screening, cf. Fig. 3(a) in the main manuscript. This has a large impact on optics, as shown in Fig. R1(a), where the photoluminescence (PL) at 30 K is plotted for the hBN-encapsulated and free-standing case (blue and red line, respectively). While the hBN-encapsulated lateral heterostructure shows

Figure R1: **Environment-dependent charge transfer (CT) exciton PL:** (a) Photoluminescence, (b) energy of the lowest exciton E , and (c) CT exciton binding energies X^b comparing hBN-encapsulated (blue) and free-standing lateral heterostructures (red). In (b,c) we consider different band offsets ΔE_v , with the blue area indicating when CT excitons are the lowest-energy state under hBN-encapsulation but not for the free-standing case.

a clear low-energy feature X_{CT} stemming from CT excitons (blue), only the monolayer peak X_{Mo} is visible in the free-standing case (red).

Besides a better fundamental understanding of exciton optics in lateral heterostructures, our work indicates a number of possible technological applications in optoelectronics, as confirmed by the referee. The mentioned 1D confinement could be realized due to the large energy separation between the monolayer and CT excitons. Furthermore, an efficient charge separation becomes realistic due to the small binding energy of CT excitons (cf. Fig. 3(b) in the main manuscript). Importantly, our work predicts a large energy separation and a small binding energy for typical band-offsets of approximately 200 meV [APL **108**, 233104 (2016)], cf. Figs. R1(b-c). The small binding energies go well beyond previous studies of lateral [15] or vertical heterostructures [Nano Lett. **17**, 938 (2017)]. The CT binding energy is about one order of magnitude larger for the free-standing case than for hBN-encapsulation, cf. Fig. R1(c). This can be explained by a more efficient screening of the Coulomb interaction in hBN-encapsulated samples, resulting in a weaker attraction of Coulomb-bound electrons and hole. As a direct consequence, under hBN encapsulation CT excitons have a one-order-of-magnitude larger dipole moment compared with interlayer excitons in vertical heterostructures, cf. Fig. 3(c) in the main manuscript. This larger dipole in turn drastically decreases the binding energy, see Fig. R3 below and Ref. [Nano Lett. **17**, 938 (2017)] for a comparison to vertical heterostructures.

In a nutshell, our study provides a theoretical prediction and experimental confirmation for the existence of stable CT excitons in lateral heterostructures. In addition, our work predicts that under hBN encapsulation these CT excitons have binding energies of just a few tens of meV and extraordinarily large dipole moments, making them interesting for many optoelectronic applications that could benefit from the efficient charge separation, fast diffusion at high-densities, and electrical control of charge transport. In the revised manuscript, we have added further comments and discus-

Figure R2: **Optical measurement of CT excitons** (a) PL linescan across the MoSe₂/WSe₂ heterojunction. The position of the heterojunction is indicated by a black dashed line. The step size is 170 nm and the excitation wavelength 633 nm. Neutral excitons of WSe₂, MoSe₂ are indicated by black arrows, while the CT exciton is denoted by a white box. (b) Power dependence of CT exciton fitted by a Lorentzian. A grey arrow shows a slight blue shift with increasing excitation power. (c) Peak position (black) and PL integrated intensity (red) as a function of the excitation power. Between 5 W and 25 W, we observe a blue-shift of approx. 1.5meV. A power law of 1.1 suggests linear PL intensity dependence within this range of excitation powers.

sions to underline the novelty and potential technological importance of our work. We have also presented new calculations comparing different substrates and added a new section and a new figure (Fig. R1) to the supplementary material.

2 Comment ‘Another critical point is the agreement between experiment and theory reported in fig. 4. Specifically, the PL peak at -100meV, ascribed to interface exciton, exhibits a bandwidth larger than the theoretical prediction and a different peak position for the two junctions (fig. 4 e,f). Moreover, the defects induce PL peaks below the absorption edge (DOI: 10.1021/acsaem.8b01561) and their appearances are most visible at low temperature (DOI: 10.1063/1.4729920). Assuming an inhomogeneous spatial organization in the lateral heterostructure, the spatially localized PL peak (ascribed in the manuscript to an interface exciton) can be explained by a defect level. For instance, a similar peak around 1.45 eV is also observed in fig. 2b of ref. [14], but it cannot be ascribed to interface exciton because it is not localized in the lateral heterojunctions. The authors should validate the experiment/theory matching ruling out other easier explanations of PL peak. ’

Answer The referee raises again an important point. We ascribe the broader shape of the measured peak to a combination of effects that occur at the interface and that have not been considered in our microscopic theory. We note that the shape and energy of the observed peak varies between different heterojunctions most likely due to a combination of strain and local disorder (either dielectric or related to the intrinsic crystal). The difference in the lattice mismatch that can generate strain, the presence of a finite junction width, as well as possible impurities trapped in the heterojunction result in variations along the interface affecting the energy of the excitonic resonances and broadens the measured peak. We agree with the referee that the low-energy peak could be in principle due to defects, however we expect defect-induced peaks to be lower in energy. In our measurements, we statistically observe an additional peak around 1.45 eV, i.e. 200 meV below X_{Mo} that we ascribe to defects, in accordance with theoretical studies predicting peaks approximately 250 meV below the bright one in MoSe₂ monolayers [PRL 121, 167402 (2018)]. In contrast, the CT exciton is much closer to the monolayer exciton and only appears at the interface.

To further examine the nature of the observed peak and exclude its link with de-

fect states we have performed additional experiments measuring PL linescans across MoSe₂/WSe₂ heterojunctions at T=4K (cf. Fig. R2). At 1.525eV a peak with a linewidth of approx. 5 meV appears only at the interface (Fig. R2(a)). To exclude its possible origin due to defect states we perform power dependent experiments and fit the integrated intensity with a power law function ($y = ax^\beta$). We measure a close to linear behavior as a function of power ($\beta=1.1$, red points in Fig. R2(c)) contrary to the sub-linear saturating behavior expected from defect states (cf. new Refs. [7-10] in the supplementary material). In the same power range, neutral excitons of MoSe₂ and WSe₂ also show a linear dependence.

In addition, we plot the emission energy as a function of the excitation power (Fig. R2(c), black points). We find a slight blue shift for the increasing excitation power that is most likely due to the dipolar Coulomb repulsion between CT excitons exhibiting a permanent electric dipole moment. This observation is similar to the behaviour of interlayer excitons in vertical heterostructures, cf. Refs. [10-14] in the supplementary material, and makes a connection of this peak to defect states highly unlikely. The measured blue-shift of the low-energy peak could thus indicate its dipolar origin and further confirm our assignment to CT excitons.

In the revised manuscript and new supplementary material, we have commented and presented new power-dependent studies to rule out defects as explanation for the observed low-energy peak. We have added Fig. R2 to the supplementary material together with a challenging description. Furthermore, we have added a new paragraph discussing the role of strain and defects at the interface of the lateral heterostructure.

Reviewer: 2

Comment *“In this study, Rosati and colleagues presented a joint theory-experiment study to investigate the bound charge-transfer excitons at the interface of lateral two-dimensional heterostructures. With a comprehensive calculations, they discussed how interface- and dielectric effects can modulate the exciton energy and binding energy of the charge-transfer exciton, and propose that the optimal conditions for an experimental observation of charge transfer excitons are the narrow junction widths (in the range of a few nm), relatively large band offsets (above 100 meV), and an intermediate dielectric screening ($\epsilon \approx 2 - 5$). Following the theoretical prediction, they observed a PL emission about 80 to 100 meV below the PL peak of MoSe₂ resonance in hBN-encapsulated MoSe₂-WSe₂ junctions, which is assigned as the charge-transfer exciton. These results sound reasonable and maybe useful for the further theoretical and experimental studies on lateral two-dimensional heterostructures. ”*

Answer We thank the referee for acknowledging that our results are interesting and could trigger further theoretical and experimental studies in the still relatively young research field of lateral heterostructures. We also appreciate the valuable and constructive comments that helped us improve the presentation of our work.

1 Comment *“ In Figure 2, the calculated binding energy of charge-transfer exciton is only about 30 meV in hBN-encapsulated MoSe₂-WSe₂ lateral heterostructure. While the binding energy of free standing TMD monolayers can be a few hundred meV in the previous studies (Ref.15 and GW-BSE calculations). Although I understand the dielectric screening can decrease the binding energy, the value presented here is too small. Maybe more comparisons with the previous experimental and theoretical results are needed.”*

Figure R3: **Charge transfer exciton binding energy.** Binding energy X_{CT}^b of the CT exciton as a function of its dipole d_{e-h} (for a fixed junction width of $w = 2.4$ nm and a band offset of $\Delta E_v = 215$ meV). The vertical red/blue lines indicate the binding energy for the free-standing case and hBN encapsulation, respectively.

Answer We thank the referee for raising this important point. The low binding energies X_{CT}^b of charge transfer excitons in the investigated hBN-encapsulated lateral heterostructures are indeed one of the main messages of our work and make them highly interesting materials for optoelectronic devices, as discussed in the response to the first comment of the first referee.

We follow the advice of the referee and compare our results to previous studies: For monolayer excitons, we find under hBN encapsulation a binding energy of almost 150 meV (orange line in Fig. 3(b) of the main manuscript), similarly to previous studies (e.g. PRB **99**, 205420 (2019)). For the free-standing case, we find more than 400 meV, again in accordance with previous studies [15], and as expected by the referee. To understand the drastic decrease of the CT exciton binding energy down to few tens of meV in hBN-encapsulated lateral heterostructures, we need to consider more than the simple increase of the dielectric screening.

In Fig. R3, we investigate the relation between the dipole and the binding energy by exploiting its dependence on the dielectric constant from Fig. 3 in the main manuscript. For increasing dipoles d_{e-h} , we observe a monotonic decrease of the binding energy, qualitatively similar to the one reported in Ref. Nano Lett. **17**, 938 (2017) for suspended vertical MoS₂-WSe₂ heterostructure with increasing interlayer separation. This indicates that an increased electron-hole separation reduces the binding energy. CT excitons, however, show a much larger dipole than interlayer excitons, because in the latter the separation is limited to the interlayer distance. As a result, the large dipole of CT excitons for hBN-encapsulated lateral heterostructures is responsible for their strongly reduced binding energy of few tens meV. Considering substrates with reduced dielectric screening, the stronger Coulomb interaction keeps the electrons and holes together, leading to dipoles $d_{e-h} \approx 1$ nm comparable with vertical heterostructures [Nano Lett. **17**, 938 (2017)]. In this free-standing case, we predict binding energies in the range of 200 meV, in agreement to the ones found in vertical MoS₂-WSe₂ heterostructures with comparable layer separation [Nano Lett. **17**, 938 (2017)].

In the revised manuscript we have added a new paragraph discussing the origin of the small binding energy of CT excitons and providing a comparison to binding energies in TMD monolayers and vertical heterostructures. Furthermore, in the new supplementary material, we have added a new figure (Fig. R3) discussing the relation between the binding energy and the dipole of CT excitons.

Figure R4: **Energy landscape in hBN-encapsulated MoSe₂-WSe₂ heterostructures.** (a) Spatial variation of the minimal excitonic energies for the three key exciton states $K_i K_i$, $K_i K'_i$ and $K_i \Lambda_i$, with $i=Mo, W$ distinguishing between different valleys in the MoSe₂ and WSe₂ layer, respectively. While the momentum-dark excitons ($K_W K'_W$ and $K_W \Lambda_W$) are the energetically lowest states in the WSe₂ monolayer, they have a higher energy than the bright exciton $K_{Mo} K_{Mo}$ in the MoSe₂ layer due to the larger single particle band-gap in WSe₂. Single-particle energy alignment and free-CT energy E_{CT}^0 for the configuration with a hole in the K valley of the (b) WSe₂ and (c) MoSe₂ layer. Due to the specific alignment, $K_W K_{Mo}$ is the CT exciton with the minimal energy.

2 Comment “In Figure 3, the calculate CT exciton dipole d_{e-h} can reach 16 nm, which is larger than the adopted interface width $w=2.4$ nm, can the author explains why?”

Answer The referee addresses again an important question. In general, a decreased binding energy is always associated with a larger electron-hole separation. The measured root-mean-square radius in hBN-encapsulated WS₂ monolayers increases by a factor of approximately 3 when going from 1s to 2s states [Phys. Rev. B 98, 075438 (2018)], reflecting the smaller binding energy. Similarly for CT excitons in lateral heterostructures, the increase of the dielectric screening in the case of hBN-encapsulation induces a decrease of the binding energy, which in turn triggers a drastic increase of the dipole that can be much larger than the junction width. This insight is new compared to vertical heterostructures, where the dipole is limited to the fixed interlayer separation. In the revised manuscript, we have added a comment and new references on the interplay of the weak exciton binding energy and the increased electron-hole separation.

3 Comment “In the previous experimental studies [Nano letters, 2015, 15(10): 6494-6500.], the WSe₂ monolayer exhibits an indirect quasi-particle gap with the conduction band minimum located at the Q-valley. While, in this study, the author assumes the conduction band minimum of WSe₂ is at K valley and take MoSe₂-WSe₂ as type II heterostructure. So is it possible that MoSe₂-WSe₂ is Type I lateral heterostructure, and that the observed PL emission below the PL peak of MoSe₂ resonance is from the WSe₂ indirect exciton, instead of the charge-transfer exciton?”

Answer This is a highly relevant and important question. In WSe₂ monolayers, the indirect or

momentum-dark KK' and KA excitons (also called KQ), formed with a hole in the K valley and an electron in the K' and Λ valley, respectively, are energetically below the bright KK exciton, resulting e.g. in phonon sidebands at cryogenic temperatures, cf. e.g. Nano Lett. 20, 2849 (2020). To illustrate this, in Fig. R4(a) we plot the minimal energy of dark excitons (yellow and green thin lines for KK' and KA , respectively) as a function of space along the lateral heterostructure. These values are compared to the energy of monolayer as well as CT bright excitons (thick lines). All valleys have an additional sub-index $i = \text{Mo, W}$ to distinguish between bands in the MoSe_2 and WSe_2 layer, respectively. The spatial variations of the minimal energies are obtained microscopically by calculating the energy separation between bright and dark excitons in the monolayer case (by solving the Wannier equation) and by taking a spatial variation of the KK energy, as described in the methods section in the main paper. At the WSe_2 side the dark excitons are the energetically lowest states, as reflected e.g. by the phonon sidebands at cryogenic temperatures (cf. e.g. Nano Lett. 20, 2849 (2020)). However, their energy is still larger than the bright exciton on the MoSe_2 side, because the energy separation $X_W - X_{\text{Mo}}$ between WSe_2 and MoSe_2 monolayer peaks is larger than the bright/dark exciton separation in WSe_2 , cf. Fig. R4(a).

In Figs. R4(b,c), we discuss also other possible CT-exciton configurations by considering the single-particle band alignment. Lateral MoSe_2 - WSe_2 heterostructures are expected to have a type II alignment, with the conduction band minimum located in the MoSe_2 layer [APL 108, 233104 (2016)]. In Fig. R4(b), we consider CT excitons formed with the hole in the K valley of WSe_2 and electrons from the K or the Λ valley in MoSe_2 . Due to the single-particle energy dispersion of MoSe_2 [2D Mater. 2, 022001 (2015)], the free-CT energy E_{CT}^0 of the $K_W K_{\text{Mo}}$ excitons is clearly smaller than the one of $K_W \Lambda_{\text{Mo}}$ excitons. The energy difference correspond to $\Delta E_{\Lambda}^{\text{Mo}}$, which is the separation between the Λ and the K valley in MoSe_2 ($\Delta E_{\Lambda}^{\text{Mo}} \approx 160$ meV according to the Ref. [2D Mater. 2, 022001 (2015)]). In Fig. R4(c), we consider CT exciton with the hole located in the MoSe_2 layer. These have energies much larger than E_{CT}^0 . As a result, the CT excitons $K_W K_{\text{Mo}}$ considered in our work are much smaller than the energies of all other CT configurations. The situation would be different in lateral heterostructures, where tungsten-based materials have the minimum conduction band. Given the band alignments between different TMD monolayers [APL 108, 233104 (2016)], this could potentially be the case for WSe_2 - WS_2 heterostructures, where we could expect the ground state to be the CT exciton with the electron lying in the K' or Λ -valley in the WS_2 layer.

In the revised manuscript, we have added a new paragraph (including new references) on the excitonic band alignment for the considered MoSe_2 - WSe_2 lateral heterostructure as well as a brief statement on the expected energy landscape in other heterostructures. Furthermore, we have added the new Fig. R4 to the supplementary material.

Reviewer: 3

Comment *“In the manuscript titled “Interface engineering of charge-transfer excitons in 2D lateral heterostructures”, the authors report on an in-depth investigation of the electronic properties of charge transfer excitons, which are supported at the interfaces of lateral transition metal dichalcogenide (TMD) heterostructures. Due to the type II band alignment, these excitons are spatially indirect, alike their interlayer counterparts in vertical TMD heterostructures. Unlike the latter, however, the number of works available in the literature is comparatively small. The theoretical works focus mainly on the band alignment between the constituent materials by making use of ab-initio methods, while*

experimental works typically employ scanning probe microscopy. Those which use spectroscopic tools to unveil the properties of these heterostructures mainly do so at room temperature, presumably due to the relatively poor optical quality of the materials, usually grown by chemical vapour deposition. In this manuscript, the authors propose for the first time a fully microscopic theoretical investigation of the charge transfer exciton, they thoroughly explore their properties by (theoretically) performing dielectric and interface engineering. The manuscript is complemented by an interesting comparison between the simulated and experimental PL spectra, in which the authors identify a peak at energy lower than that of MoSe₂ exciton with the charge transfer exciton. This represents the first time this excitonic complex has been observed experimentally.

The manuscript is very well written. The authors guide the reader through their results very clearly. Both this and the topic make this manuscript of potential interest for the broad readership of Nature Communications.

”

Answer We thank the referee for the overall positive assessment and in particular for acknowledging that our work is suitable for the broad readership of Nature Communications. We also appreciate the valuable comments that helped us improve the presentation of our work.

1 Comment “In the “interface engineering” section, the authors discuss in detail how a few key quantities (e.g., energy and dipole of the charge transfer exciton). They however do not devote as much attention to the dependence of the binding energy on the band offset. Their simulation summarized in Fig 2f show that this quantity decreases for increasing band offset, which might be counterintuitive. I suggest to comment on this in the revised manuscript.”

Answer We thank the reviewer for addressing this important point. In Figs. R1(b,c) we show two important quantities as a function of the band offset ΔE_v , i.e. the energy separation between the energetically lowest state and the monolayer exciton X_{Mo} as well as the CT exciton binding energy. For small offsets, the energetically lowest state is the monolayer exciton, i.e. $E \equiv X_{\text{Mo}}$. For $\Delta E_v \gtrsim 110$ meV (200 meV), the CT exciton becomes the ground state under hBN-encapsulation (free-standing). After this threshold, the energy separation $E - X_{\text{Mo}}$ between CT and MoSe₂ exciton increases for increasing offsets, reflecting a decrease of the CT bandgap E_{CT}^0 , cf. Fig. 1 in the main manuscript. In Fig. R1(c), we study the CT-exciton binding energy X_{CT}^{b} for offsets above the threshold. Increasing the offset, X_{CT}^{b} initially decreases before saturating. The decrease is likely induced by an interplay between CT and monolayer excitons, when they are energetically close.

In the revised manuscript, we have added a new paragraph on the variation of the charge transfer exciton energy and the CT binding energy with increasing band offsets. In the new supplementary material, we have included a new figure (Fig. R1) with a discussion.

2 Comment “Can the author comment on the comparison of their simulated value of the binding energy of the suspended heterostructure (epsilon=1) with that reported in PRB 98 115427, which seems to be considerably higher?”

Answer Here, it is important to take into account the used substrate. In Fig. SR1(c), we show the CT binding energies for hBN-encapsulated lateral heterostructures (blue) and for the free-standing case. In the latter, we find a binding energy close to 200 meV - in excellent agreement with PRB **98**, 115427 (2018), where free-standing heterostructures have been considered. Besides the quantitative match with Ref. [PRB **98**, 115427 (2018)], also the qualitative behaviour for increasing band offsets agrees well.

In the revised manuscript, we have emphasized the important role of substrate-induced dielectric screening for the CT exciton binding energy.

3+4 Comment *“In the simulated spatial-dependent PL spectrum of fig 4c, the intensity of the charge transfer exciton seems to be larger on the WSe2 side: could the authors clarify why this is the case?”*

At low temperature, always in fig 4c, one can note that a non-zero signal related to the exciton of MoSe2 is present also relatively far from the interface: is this related to diffusion effects?”

Answer The referee raises two important points that both can be traced back to the different energy of MoSe₂ and WSe₂ excitons, i.e. $X_W \gg X_{Mo}$. At higher temperatures, both X_{Mo} and X_W have a finite signal for excitations slightly on the WSe₂ and MoSe₂ side of the junction, respectively. This is due to the finite spatial size of the laser pulse, i.e. that we partially excite both layers when the laser pulse is close to the junction. Furthermore, at low temperatures, the predicted signal is not symmetric anymore, with X_{Mo} and X_{CT} being much more intense than X_W and extending deep into the WSe₂ monolayer region. We can ascribe this to an interplay of the center-of-mass wavefunction and the state-occupation. At small temperatures, one has $X_W - X_{Mo} \gg k_B T$, resulting in a much higher occupation of the MoSe₂ bright exciton. Thus, even a small wavefunction overlap can result in a considerable signal on the WSe₂ side of the junction. Similar arguments hold for the intensity of the charge transfer exciton X_{CT} , which is more intense compared to X_W than to X_{Mo} reflecting the larger energy separation.

In the revised manuscript, we have added a new paragraph discussing the intensity and the spatial extension of the peaks in the PL spectrum.

5 Comment *“Did the authors perform some spatial dependence PL measurements at low temperature? In a prior publication, they made use of tip enhanced PL, which I suppose would not be possible to perform at cryogenic temperature, but perhaps a raster scan of the PL spot across the heterostructure could be interesting and show also experimentally the simulated spatial dependence of the charge transfer exciton.”*

Answer We are grateful for this comment and for the careful examination of our work. Indeed, in our previous study we performed tip-enhanced PL experiments, and we confirm that such experiments are not possible at low temperatures. The point and the idea of the reviewer on performing PL spatial maps is very helpful. We have performed additional experiments measuring PL linescans across MoSe₂/WSe₂ heterojunctions at $T=4K$ (cf. Fig. R2). At 1.525eV a peak with a linewidth of approx. 5 meV appears only at the interface (Fig. R2(a)). To exclude its possible origin due to defect states we perform power dependent experiments and fit the integrated intensity with a power law function ($y = ax^\beta$). We measure a close to linear behavior as a function of power ($\beta=1.1$, red points in Fig. R2(c)) contrary to the sub-linear behavior expected from defect states (see Refs. [7-10] in the supplementary material). In the same power range, neutral excitons of MoSe₂ and WSe₂ also show linear dependence.

In addition, we plot the emission energy as a function of excitation power (Fig. R2(c), black points). We find a slight blue shift with the increasing excitation power that is most likely due to dipolar Coulomb repulsion between CT excitons with a permanent electric dipole moment. This observation is similar to interlayer excitons in vertical heterostructures, cf. Refs. [10-14] in the supplementary material, and makes a connection of this peak to defect states highly unlikely.

Following the last remark from the reviewer, we emphasize that the PL linescan indicates the brighter CT exciton emission possibly appearing at the WSe₂ side, in agreement with the simulated spatial dependence (Fig. R2(a)). However, we need

to be cautious here due to uncertainties in the precise nanometer-size heterojunction compared to a μm size laser spot.

In the supplementary material, we have added Fig. R2 including a corresponding section.

6 Comment *“How does the ratio of intensity between the (neutral) exciton of MoSe2 and the charge transfer exciton compare in the simulated and experimental spectra?”*

Answer The predicted and measured intensities shown in Fig. 4 of the paper are in a good agreement, exhibiting a ratio $X_{\text{CT}}/X_{\text{Mo}} \approx 0.2 - 0.3$ for both theory and experiment. We should, however, note that absolute values of intensities should be treated with care as their calculation and measurement are difficult. From the theoretical point of view, we are taking a local thermalization approximation, while transient features could be present. From the experimental point of view, local disorder could be present, as reflected e.g. by the larger energetic broadening of the CT peak compared to the monolayer resonances.

In the revised manuscript, we have added a comment about the intensity ratio of different peaks in the PL spectrum.

Reviewers' Comments:

Reviewer #1:

Remarks to the Author:

The manuscript can now be published in Nature Communications

Reviewer #2:

Remarks to the Author:

I think the authors addressed my concerns, and I recommend the publication as it is now.

Reviewer #3:

Remarks to the Author:

In the resubmitted version of the manuscript, the authors addressed all concerns raised by me and, as far as I can see, by the other reviewers. I recommend to publish the manuscript in its current form.